# Caregivers’ Burden of School-Aged Children with Neurodevelopmental Disorders: Implications for Family-Centred Care

**DOI:** 10.3390/brainsci11070875

**Published:** 2021-06-30

**Authors:** Giulia Purpura, Luca Tagliabue, Stefania Petri, Francesco Cerroni, Andrea Mazzarini, Renata Nacinovich

**Affiliations:** 1School of Medicine and Surgery, University of Milano Bicocca, 20900 Monza, Italy; renata.nacinovich@unimib.it; 2Child and Adolescent Health Department, S. Gerardo Hospital, ASST of Monza, 20900 Monza, Italy; luca.tagliabue@unimib.it; 3Department of Developmental Neuroscience, IRCCS Stella Maris Foundation, 56128 Pisa, Italy; s.petri@fsm.unipi.it; 4Clinic of Child and Adolescent Neuropsychiatry, Università degli Studi della Campania “Luigi Vanvitelli”, 80100 Naples, Italy; francesco.cerroni@unicampania.it; 5Clinic of Rehabilitation “A Ruota Libera”, 00069 Trevignano Romano, Italy; andrea.mazzarini@libero.it

**Keywords:** neurodevelopmental disorders, caregivers, burden

## Abstract

Caregivers of children with neurodevelopmental disorders play a central role during the rehabilitation and education processes, but they have an increasing risk of psychosocial problems even if the literature is not so agreed upon the specific and predisposing factors to that. The aim of this study was to examine possibly differences of burden levels in an Italian sample of principal caregivers of children with different kinds of neurodevelopmental disorders and to investigate the possible links between some clinical and sociodemographic variables and the levels of caregiver’s burden. 105 caregivers of school-aged children with neurodevelopmental disorders were included in the study and completed three online questionnaires (General Questionnaire, Caregiver Burden Inventory, Zarit Caregiver Burden Scale). Results highlighted that about the half of caregivers show from moderate to high levels of stress, but parents of children with Autism Spectrum Disorder and Intellectual Disability show greater difficulties than parents of children with Attention-Deficit Hyperactivity Disorder, Language and/or Learning Disorder, and Developmental Coordination Disorder. Moreover, it was evident a negative correlation between the burden levels and the age of children, but also a direct correlation between the burden levels and the weekly hours of rehabilitation. These findings show that severity of caregiver’s burden is dependent by the type of neurodevelopmental disorder and suggest that an ecological and family-centred approach is necessary to guarantee the life health developmental course of these children.

## 1. Introduction

Neurodevelopmental disorders are a group of conditions with early onset in life of affected individuals, characterized by developmental deficits and behavioural impairments in several areas of functioning of daily life (personal, social, academic, and occupational domains) [1]. According to the Diagnostic and Statistical Manual of Mental Disorders—Fifth Edition (DSM-5), the advances in clinical neuropsychiatry permit to classify the brain dysfunctions of the developmental age, emphasizing the approach to clinical and complex cases and highlighting the pervasive and cascading effect that the early onset of these disorders can have on the evolution of individuals [1,2].

Within the section of neurodevelopmental disorders of DSM-5, intellectual disability, autism spectrum disorders, communication disorders, specific learning disorders, attention-deficit hyperactive disorders and motor disorders were included [2]. Despite this precise categorization, it is known by scientific literature that the complexity and heterogeneity of neurodevelopmental disorders is given both by the intertwining of the components of development (perceptual, motor, cognitive, linguistic, social, and emotional features) and by the often present numerous comorbidities, and these factors can make extremely difficult the adaptation to the environment with substantial limitations in autonomy and restriction of daily activities, thus requiring continuous assistance [3,4,5]. For these reasons, caregivers play a central role during the rehabilitation and education processes of these children, so their well-being and health are very dependent also by the whole health condition of their family in terms of participation to social life, continuity of healthcare as well as education, social and community services and supports [6].

Health development of the entire family of children with neurodevelopmental disorders must be considered as an adaptive process, with the principal aim of promoting resilience and plasticity in the face of changing and often constraining environmental contexts [6,7]. This issue is confirmed also by the fact that high stress levels and impaired physical/mental health in parents or caregivers can restrict opportunities for early and effective intervention and can further compromised the life health developmental course of these children. In fact, caregivers have an increasing risk of psychosocial problems even if the literature is not so agreed upon the specific and predisposing factors to that [4,8,9,10,11].

In a recent metanalysis, Masefield and colleagues [8] confirmed that mothers of young children with developmental disabilities may have poorer health than those with typically developing children, but also suggested that clinical research is still needed to identify whether the relationship is causal and, if so, social, educational, and rehabilitative interventions could reduce the negative effect of caregiving. Consequently, understanding the stress and burden levels of caregivers of disabled children is very important during the rehabilitation process and can have several implications for improving psychophysical well-being of the child and the family.

The purpose of this study was to examine possibly differences of burden levels in an Italian sample of principal caregivers of children with different kinds of neurodevelopmental disorders and to investigate the possible links between some clinical and sociodemographic variables and the levels of caregiver burden.

## 2. Materials and Methods

### 2.1. Sampling and Data Collection

This observational study was conducted by a group of clinicians and researchers in the Department of Medicine and Surgery of the University of Milano Bicocca (Italy), specifically settled in the Child Neuropsychiatry Unit of the San Gerardo Hospital of Monza (Italy). Data collection was performed with the voluntary help of several clinicians of the South, Centre, and North of Italy. The research was conducted on a sample of 105 voluntary caregivers (89 mothers; 15 fathers; 1 other) related to consecutively attending outpatients with a previously established primary diagnosis of neurodevelopmental disorders according to DSM-5 criteria. The inclusion criteria for the study were as follows; (i) principal caregivers of children with neurodevelopmental disorders assessed by a multidisciplinary team; (ii) age of children between 6 and 18 years; (iii) availability of devices and e-mails to complete the questionnaires.

Specifically, the patient had to have a primary diagnosis of Autism Spectrum Disorder or Intellectual Disability or Attention-Deficit Hyperactivity Disorders or Communication Disorders or Specific Learning Disorders or Developmental Coordination Disorder and the relative had to be the principal caregiver. The caregiver was approached separately by the rehabilitation’s professional of his/her child (neurodevelopmental disorders therapist or speech therapist or motor therapist or psychologist). After this approach and the first oral explanation of the study, caregiver received an e-mail with the “Internet Link” that permitted to complete the self-report online.

In the first part of form, it was reported the aim of the study and some clinical and sociodemographic information were requested, indeed in the second part, two questionnaires regarding the stress and the burden levels of caregivers were proposed (see Measures Section). The entire compilation was completely anonymous. Recruitment was performed from the 9 March 2021 to the 5 May 2021.

### 2.2. Measures

#### 2.2.1. General Questionnaire

The General Questionnaire (GQ) permitted the collection of data about some sociodemographic information: provenience (region of Italy), ages of the child and caregiver, type of relationship with the child, primary diagnosis, attended school class by the child, instruction level of caregiver, employment status of caregiver, gestational age at birth of the child, presence of epilepsy in the child, and weekly hours of rehabilitative interventions of the child. Data about GQ are reported in Table 1 and Table 2.

#### 2.2.2. Caregiver Burden Inventory

Caregiver Burden Inventory (CBI) is a multidimensional questionnaire [12] that consists of 24 items on 0–4 Likert-type scales (from 0 that is “not at all disruptive” to 4 that is “very disruptive”), in our study proposed in the Italian Version by Marvardi and collaborators [13]. The items are organized in 5 subscales to evaluate 5 different dimensions of caregiver burden: CBI-time dependence (items from 1 to 5), which measures the burden associated with the caregiver’s time restriction; CBI-developmental (items from 6 to 10), which evaluates the caregivers’ feeling of being “off-time” in their development compared to their peers; CBI-physical (items from 11 to 14), which allows the description of physical effort and physical problems consequent of caregiving; CBI-social (items from 15 to 19), which assesses the impact on interpersonal and social relationships and the perception of a role conflict; CBI-emotional (items from 20 to 24), which evaluates feelings of shame and embarrassment regarding the patient.

The total score (CBI–tot) ranges from 0 to 96, with higher scores indicating greater feelings of burden (total scores between 0 and 24 indicate low risk of burnout, scores between 25 and 36 moderate risk of burnout, and scores above 36 high risk), while the subtotal scores about the 5 dimensions rages from 0 to 20, except for CBI-physical that ranges from 0 to 16.

The internal reliability of the Italian-language version has been validated by Marvardi et al. [13] but it was confirmed with our sample also in this study (Cronbach alpha = 0.95).

#### 2.2.3. Zarit Caregiver Burden Scale

Zarit Caregiver Burden Scale (ZCBS) is a self-report questionnaire, that consists of 22 items on 0-4 Likert-type scale (from 0 that is “never” to 4 that is “almost always”), to determine the effect of caregiving on the life of the individual concerned [14]. In our study the Italian Version by Chattat and collaborators was administered [15]. The ZCBS measures different issues related to quality of life, psychological suffering, financial difficulties, shame, guilt in social and family relationships. The ZCBS items are sorted into three scales: Personal Strain Scale (items 1, 4, 5, 6, 9, 13, 14, 16, 18, and 19); Role Strain Scale (items 2, 3, 7, 8, 10, 11, 12, 15, 17, and 22); and Guilty Scale (items 20, 21) [16]. Considering the entire ZCBS items, scores below 20 indicate little or no burden, between 21 and 40 a mild to moderate burden, between 41 and 60 a moderate to severe burden and between 61 and 88 a severe burden [15]. The internal reliability of the Italian-language version has been validated by Chattat et al. [15] but it was confirmed with our sample also in this study (Cronbach alpha = 0.93).

### 2.3. Statistical Analysis

Study data were analysed on SPSS 16.0 software. To examine differences in caregivers’ behaviours, the total sample was divided into 5 subgroups: Group 1, caregivers of children with Autism Spectrum Disorders (ASD); Group 2, caregivers of children with Intellectual Disability (ID); Group 3, caregivers of children with Attention-Deficit Hyperactivity Disorder (ADHD); Group 4, caregivers of children with Language and/or Specific Learning Disorders (LD); Group 5, caregivers of children with Developmental Coordination Disorders (DCD). Descriptive statistics are reported where appropriate.

To highlight caregiver’s burden of children with different neurodevelopmental disorders and investigate the differences between the 5 subgroups, measures of global score and partial scores included in the CBI and ZCBS were considered, and a Univariate ANOVA was carried out. Post-hoc comparison was done by means of a Bonferroni test. A *p*-value below 0.05 was interpreted as significant.

Correlation between the two burden indices was verified. Successively, two-tailed Pearson correlation analysis was used to determine the type and power of correlations between caregiver burden (ZCBS and CBI, total and partial scores) and some sociodemographic data of the entire sample (age of patients, age of caregivers, instruction level of caregivers). Finally, two-tailed partial correlation analysis, controlling for age of children and provenience, were performed to verify the relationship between caregiver’s burden levels and (i) employment status of caregivers, (ii) subgroup of diagnosis, (iii) gestational age at birth, (iv) presence of epilepsy and (v) weekly hours of rehabilitation.

## 3. Results

### 3.1. Characteristics of the Sample

Principal characteristics of the sample are shown in Table 1 and Table 2. For this study, 105 carers of Italian patients with neurodevelopmental disorder were recruited (North of Italy: 23.8%; Centre of Italy: 45.7%; South of Italy: 30.5%). Of these carers, 89 were mothers (84.7%), 15 were fathers (14.2%) and 1 other relative (1.1%). The mean age of caregivers was 41.93 years (SD = 5.17), ranging from 31 to 57 years of age. Most of the caregivers had a high school education (57.1%), while the 13.3% had a primary or secondary level of education and 29.5% had a university level of education. Also, the 61.1% of caregivers were employed, while the 38.9% of caregivers were unemployed or homemaker. Regarding sociodemographic data of children with neurodevelopmental disorders, their mean age was 8.95 years (SD = 2.5), ranging from 6.1 to 15.8 years of age. The 8.6% of children attended the last year of kindergarten, the 74.3% one attended the primary school, the 13.3% one attended the secondary school and finally the 3.8% one attended the high school. As regards siblings of children, the 27.6% of children had not siblings, while the 59% one had one sibling and the 13.4% one had 2 or more siblings.

Primary diagnoses were ASD (32.4%), ID (14.3%), ADHD (14.3%), LD (27.6%) and DCD (11.4%). The 81% of them were born full-term and the 19% one was born preterm. Only 4 children were in pharmacotherapy for epilepsy. Finally, most of the sample (53.3%) performed a low frequency rehabilitation (from 1 to 3 h per week), while the 24.8% of children performed a moderate-high frequency rehabilitation (more than 3 h per week), and only the 21.9% of the children performed only a monthly or bimonthly follow-up.

### 3.2. Differences in Caregiver Burden between Groups

CBI and ZCBS Total and Partial Scores across the whole sample are reported in Figure 1 and in Table 3 (mean CBI-total: 26.93; SD = 21.16; mean ZBCS-total: 24.96, SD = 17.48), showing that about the half of caregivers had moderate or high risk of burnout.

Moreover, the univariate ANOVA (see Table 3) comparing total and partial scores of CBI and ZCBS of different groups (ASD, ID, ADHD, LD, DCD) indicated a significant effect of diagnosis on the scores at all investigated dimensions (CBI-tot: F(1,5) = 15.32, *p* = < 0.001; CBI-time: F(1,5) = 23.39, *p* = < 0.001; CBI-developmental: F(1,5) = 12.90, *p* = < 0.001; CBI-physical: F(1,5) = 10.62, *p* = < 0.001; CBI-social: F(1,5) = 3.46, *p* = 0.011; CBI-emotional: F(1,5) = 3.91, *p* = 0.005; ZCBS-total: F(1,5) = 11.22, *p* = < 0.001; ZCBS-Personal Strain Scale: F(1,5) = 6.26, *p* = < 0.001; ZCBS-Role Strain Scale: F(1,5) = 12.88, *p* = < 0.001; ZCBS-Guilty Scale: F(1,5) = 7.43, *p* = < 0.001). Table 3 and Figure 2 show that Bonferroni post-hoc t-test highlighted several significant statistical differences between mean scores of the ASD group and the ADHD group (CBI-tot: *p* = < 0.001; CBI-time: *p* = < 0.001; CBI-developmental: *p* = < 0.001; CBI-physical: *p* = < 0.001; ZCBS-total: *p* = 0.002; ZCBS-Role Strain Scale, *p* = < 0.001; ZCBS-Guilty Scale: *p* = 0.002), between ASD group and LD group (CBI-tot: *p* = < 0.001; CBI-time: *p* = < 0.001; CBI-developmental: *p* = < 0.001; CBI-physical: *p* = < 0.001; CBI-social: *p* = 0.019; CBI-emotional: *p* = 0.005; ZCBS-total: *p* = < 0.001; ZCBS-Personal Strain Scale: *p* = < 0.001; ZCBS-Role Strain Scale, *p* = < 0.001; ZCBS-Guilty Scale: *p* = < 0.001), and between ASD group and DCD group (CBI-tot: *p* = < 0.001; CBI-time: *p* = < 0.001; CBI-developmental: *p* = < 0.001; CBI-physical: *p* = < 0.001; ZCBS-total: *p* = 0.001; ZCBS-Personal Strain Scale: *p* = 0.019; ZCBS-Role Strain Scale, *p* = < 0.001; ZCBS-Guilty Scale: *p* = 0.016).

Moreover, significant statistical differences between mean scores of the ID group and ADHD group (CBI-tot: *p* = 0.057; CBI-time: *p* = 0.001; CBI-developmental: *p* = 0.053), between ID group and LD group (CBI-tot: *p* = 0.001; CBI-time: *p* =< 0.001; CBI-developmental: *p* = 0.016; ZCBS-total: *p* = 0.012; ZCBS-Personal Strain Scale: *p* = 0.053; ZCBS-Role Strain Scale, *p* = 0.008) and between ID group and DCD group (CBI-tot: *p* = 0.017; CBI-time: *p* = 0.001) were found. Different profiles for each group about subscales of CBI and ZCBS are reported in Figure 3.

### 3.3. Correlations

Correlation between CBI-tot and ZBCS-tot was confirmed for the whole sample (rho = 0.890; *p* = < 0.001). Considering the whole sample, results from two-tailed Pearson’s correlation test highlighted the presence of negative significant correlations between the age of children and the burden levels of caregivers specifically in CBI-tot, CBI-time and CBI-physical (CBI-tot: rho = −0.283, *p* = 0.003; CBI-time: rho = −0.463, *p* = < 0.001; CBI-physical: rho= −0.212; *p* = 0.030), that confirm the decrement of the burden with the growth of children and probably with the achievement of more autonomy of children (see Figure 4). No correlations were found between the burden levels of caregivers and the age or educational levels of caregivers.

Two-tailed partial correlation analysis, controlling for age of children and provenience (North, Centre, or South of Italy), verified the significant correlation between caregiver’s burden levels and the weekly hours of rehabilitation, specifically regarding CBI-tot (rho = 0.222, *p* = 0.024) and CBI-time (rho = 0.237, *p* = 0.016), CBI-developmental (rho = 0.245, p = 0.13), CBI-physical (rho = 0.218, *p* = 0.027), CBI-emotional (rho = 0.227, *p* = 0.021), ZCBS-total (rho = 0.254; *p* = 0.010), ZCBS-Personal Strain Scale (rho = 0.203, *p* = 0.040), ZCBS-Role Strain Scale (rho = 0.302, *p* = 0.002). Light negative correlation was found between gestational age and ZCBS-Guilty Scale (rho = −0.188, *p* = 0.057), that reveal that the less gestational age is correlated with bigger sense of guilt of caregivers.

## 4. Discussion

The purpose of this study was to investigate the perceived burden by caregivers of children with neurodevelopmental disorder and the relationship between their burden’s levels and the characteristics of these developmental disorders. To our knowledge, this is the first study jointly assessing perceived burden and the specific components of neurodevelopmental disorders, further including the comparison of these dimensions between Italian parents of children with different types of difficulties.

This approach was motivated by the complexity of neurodevelopmental disorders, that, even if have some common elements as their onset in infancy, childhood, or adolescence, are very different in terms of clinical features during school-age period and consequently in terms of required levels of assistance.

The main result of our study is that about half of the participating caregivers show from moderate to high levels of stress, but parents of children with ASD and ID show greater difficulties as effect of caregiving than parents of ADHD, LD, and DCD children. This finding is in line with results of Lach and colleagues that suggested that caregivers of children with neurodevelopmental disorders and externalizing behaviour problems showed significant burden on their physical and psychological health [10]. Our observations could be related to the particularly pervasive condition of these two types of disturbs (autism and intellectual disability), that involving the entire functioning of subject and have cascading effects on the environment adaptation of affected individuals in all developmental domains. Specifically, ASD group’s caregivers show the higher levels of burdens both in total and partial scores of CBI and ZCBS, integrating data by Baykal et al. [11] that suggested the relationship between ASD symptoms severity and caregiver depressive symptoms.

As a matter of fact, it was widely reported by scientific literature that ASD is a complex and multidimensional condition in which language and social symptoms are just the tip of the iceberg, and in which several differences in brain connectivity underlying impairments in multisensory and sensory-motor integration of these children during developmental age [17,18]. According to several authors, ASD emerges not as a higher-order cognitive deficit, but because of an impairment of primordial ability to process low level sensory, motor, and perceptual information gained through experiencing other persons since the earliest periods of life [19,20,21]. Differences in perceptual experiences could explain the tendency to perceive the world more accurately rather than modulated by prior experience. This sensory processing deficit could justify a reduced expectation than what is about to happen and consequently it could justify their atypical behaviours that are difficult to interpret by other people [22]. This issue may suggest the higher necessity to have continuous assistance from the adults during daily activities and so the high risk of burnout in this population of parents.

Moreover, Del Bianco and collaborators [23] highlighted as the perceived parental stress and maladaptive parenting strategies of ASD children might be connected to the difficulties to understand behavioural atypicality and non-verbal communication modalities of children by their parents and as the interaction between atypical communication and distress of parents likely determines a cascade effect on the parent-child dyad.

Moreover, an important point of our study is the use of these two burden scales that analyse psychophysical effect of caregiving, regardless of factors related to parenting. Although the scores on the two questionnaires are strongly correlated, their use was important to investigate all possible components that could be source of stress (for example the CBI takes into consideration also the emotional stress factors, while the ZCBS highlights much more the socio-economic factors). As a matter of fact, in scientific literature the use of these two questionnaires with caregivers of children with neurodevelopmental disorders is still very limited.

Another interesting finding from our study is that parental burden levels decrease as children grow, particularly about time limitation and care-related physical problems. This interesting aspect could be linked to two important factors. On the one hand, the growth of children can be followed, even in children with high severity of symptoms, by an increase in levels of autonomy (for example regarding personal care or autonomy during feeding), but on the other hand it can also be linked to the Italian organization of child rehabilitation services. Often, during adolescence, the weekly rehabilitation hours decrease considerably to integrate most of the interventions in collaboration with the school and community services. For these reasons, in the future, the monitoring of the parental burden with the transition to adulthood of one’s child should be taken into consideration as a critical point, because the transition from rehabilitation services for children to rehabilitation services for adulthood is not easy for these families. As a matter of fact, for patients with neurodevelopmental disorder, an ecological approach that sets intervention in a more family, community, and daily-life setting with lifelong participation being goal, may have more long-term effects [24]. In this context a life-span perspective is very important to allow the psychophysical health of these families. For these reasons, Palisano and colleagues [6,25] proposed a new biopsychosocial model, called Life Course Health Development (LCHD), that conceptualizes health development occurring through transactions between the person and environment over time. They suggested the idea to consider the social participation of disabled individuals as a central point of health services and rehabilitation processes, as well as a fundamental element physical, mental, and emotional wellbeing of them and of their caregivers [6]. According to Palisano and colleagues, this approach to neurodevelopmental disorder’s life course encourages planning for the future and promotes coordination and continuity of healthcare between paediatric and adult systems, among education, social, and community services and supports, reducing stress levels and promoting resilience within these families [6]. According to this view, it seems spontaneous to integrate this approach to that of family-centred care, which recognizes the centrality of the family in the life of the child with neurodevelopmental disorders and in which families are viewed as partners and allies for quality and safety of clinical assistance [26].

The last important finding of our study is the correlation between caregiver’s burden levels and the weekly hours of rehabilitation, suggesting that more hours of therapy, although useful for improving child performances, may be stressful for parents. Concerning that point, several authors sustain that the efficacy of intervention is not correlated to the intensity of therapies, but indeed that the involving of parents in therapy is the most important topic. For example, Rogers and collaborators [27] demonstrated as it was no differences in acquired language skills of non-verbal ASD children between a group that received 1-h weekly sessions of therapy and a group that received daily 1-h home intervention delivered by trained parents. In a more recent study, the same author found a significant positive relationship between degree of improvement in parental fidelity of implementation of ludic therapeutic strategies and increases in child social-communication and decreases in autism symptoms, suggesting the importance of developing effective interventions that can be easily used by the families of young children with neurodevelopmental disorders [28]. Also, Green and colleagues [29] reported benefits on parent-child dyadic social communication in response to parent-mediated communication-focused intervention and suggested the power of naturalistic reinforcement of behaviour at home.

Moreover, the role of parent coaching and/or the home-based interventions in children with developmental disabilities was demonstrated by several authors, suggesting interesting effects not only on child adaptation but also on parent’s stress [30,31,32,33].

## 5. Limitations

The authors are fully aware of the two main limitation of the present study. The first limit is due to its design, because we were unable to evaluate whether changes in patients’ symptom levels also modified the caregiver burden. The second limit consist in the lack of information about economic and social supports, which also probably affect caregiver burden, and that in our study was not fully evaluated. If on one hand, we are very inclined to consider very carefully these findings, on the other hand we think that the insights from this approach allow us to analyse in depth the complexity for early interventions and provide some interesting food for thought. To our knowledge, this is the first study that highlights differences in burden levels between several types of neurodevelopmental disorders.

## 6. Conclusions

The study findings showed a significant level of perceived burden in parents of children with neurodevelopmental disorders in multiple aspects of their lifestyle, relationships, and activities. Higher levels of stress are evident in caregivers of children with primary diagnosis of ASD and ID than caregivers of children with ADHD, LD, and DCD, probably for the intrinsic and pervasive characteristics of these two types of disorders, that often are also in compresence. Notably, clinicians and therapists involved in the rehabilitation process of children with neurodevelopmental disorders need to be cognizant of the mental health and caregiving burden in parents of these children and explore for a potential experience of burden in caregivers with identifiable vulnerability during clinical daily activities. Finally, these findings, in accordance with the literature, suggest the necessity of a rehabilitative and educative approach that takes into consideration the entire nuclear family and the positive relationship between the child and his/her caregiver, that promote adaptive and positive parenting strategies and to endorse the development of a blueprint for psychosocial support for parents. For these reasons, an ecological, family-centred and lifespan program of interventions could be the most adapted for this population of patients.

## Figures and Tables

**Figure 1 brainsci-11-00875-f001:**
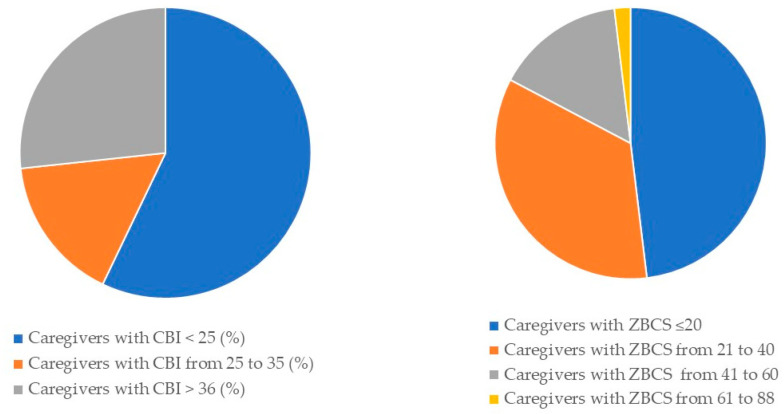
Graphic representation of percentages of Total Scores of Caregiver Burden Inventory (CBI) and Zarit Caregiver Burden Scale (ZCBS).

**Figure 2 brainsci-11-00875-f002:**
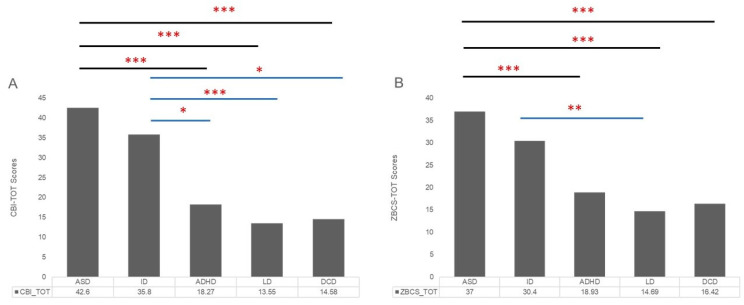
Graphic representation of means of scores ((**A**) Caregiver Burden Inventory—Total Scores; (**B**) Zarit Caregiver Burden Scale—Total Scores) in the caregiver’s groups. The asterisks indicate a significant difference between conditions: * *p* ≤ 0.05, ** *p* ≤ 0.01, *** *p* ≤ 0.005. (ASD: Autism Spectrum Disorder; ID: Intellectual Disability; ADHD: Attention-Deficit and Hyperactivity Disorder; LD: Language/Learning Disorder; DCD: Developmental Coordination Disorder).

**Figure 3 brainsci-11-00875-f003:**
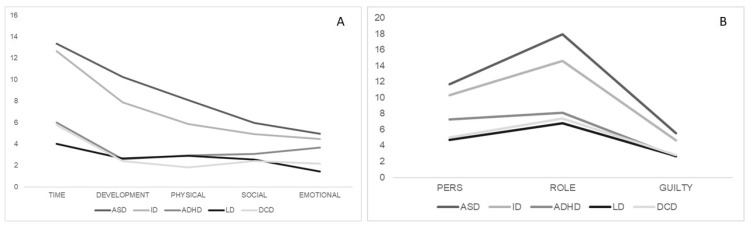
Graphic representations of different profiles of burden levels ((**A**) subscales of Caregiver Burden Inventory; (**B**) subscales of Zarit Caregiver Burden Scale) across the 5 subgroups of caregivers.

**Figure 4 brainsci-11-00875-f004:**
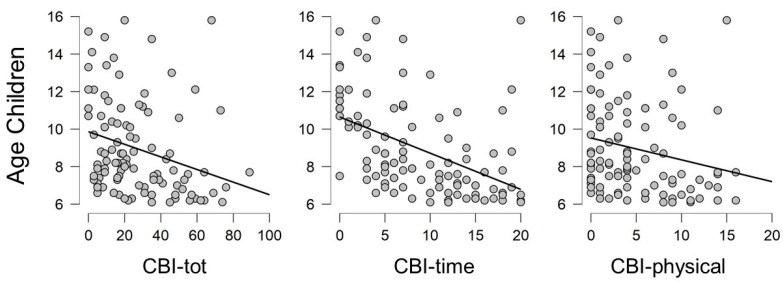
Graphic representations of negative correlations between the age of children and the level of caregiver’s burden.

**Table 1 brainsci-11-00875-t001:** Distribution and principal sociodemographic information of the sample.

Data Collection	Total	South of Italy	Centre of Italy	North of Italy
Subjects (*n*; %)	105 (100)	32 (30.5)	48 (45.7)	25(23.8)
Age of children, y,mos (mean; range)	8.9 (6.1–15.8)	8.2 (6.1–15.8)	9.8 (6.3–15.8)	8.2 (6.1–15.2)
Age of caregivers y,mos (mean; range)	41.9 (31–57)	41.4 (33–57)	42.4 (32–54)	41.6 (31–57)
Caregivers (mothers; fathers; others)	89M; 15F; 1o	27 M; 4F;1o	40M; 8F; 0o	22M; 3F; 0o
Employment status of caregivers (Yes/No)	65 Y/40 N	14 Y/18 N	33 Y/15 N	18 Y/7 N
**Primary Diagnosis**				
Autism Spectrum Disorder (*n*)	34	20	8	6
Intellectual Disability (*n*)	15	6	3	6
Attention Deficit Hyperactivity Disorder (*n*)	15	1	10	4
Language and/or Learning Specific Disorder (*n*)	29	4	18	7
Developmental Coordination Disorder (*n*)	12	1	9	2

**Table 2 brainsci-11-00875-t002:** Clinical features of the sample.

Gestational Age	Subjects (%)
Full term children (≥37 weeks)	81
Late preterm children (from 33 to 36 weeks)	14.2
Very preterm children (from 28 to 32 weeks)	1.0
Extremely preterm children (<28 weeks)	3.8
**Epilepsy**	**Subjects (%)**
No seizures	96.1
Seizures controlled by pharmacotherapy	1.9
<1 seizure/month	1
≥1 seizure/month	0
≥1 seizure/day	1
**Rehabilitation**	**Subjects (%)**
No rehabilitation, only bi/monthly follow-up	21.9
From 1 to 3 h of rehabilitation per week	53.3
More than 3 h of rehabilitation per week	24.8

**Table 3 brainsci-11-00875-t003:** Univariate ANOVA for the scores and subscores of the two questionnaires (in the 5 subgroups of caregivers).

Caregiver Burden Inventory	Mean (SD)	*p*-Value	Post-Hoc Bonferroni
CBI-Tot	26.93 (21.16)	<0.001	ASD vs. ADHD; ASD vs. LD; ASD vs. DCD; ID vs. ADHD; ID vs. LD; ID vs. DCD
CBI-time dependence	8.75 (6.02)	<0.001	ASD vs. ADHD; ASD vs. LD; ASD vs. DCD; ID vs. ADHD: ID vs. LD; ID vs. DCD
CBI-developmental	5.83 (6.10)	<0.001	ASD vs. ADHD; ASD vs. LD; ASD vs. DCD; ID vs. ADHD; ID vs. LD
CBI-physical	4.89 (4.56)	<0.001	ASD vs. ADHD; ASD vs. LD; ASD vs. DCD
CBI-social	4.06 (4.43)	0.011	ASD vs. LD
CBI-emotional	3.41 (4.01)	0.005	ASD vs. LD
**Zarit Caregiver Burden Scale**	**Mean (SD)**	***p*-Value**	**Post-Hoc Bonferroni**
ZBCS-Tot	24.96 (17.48)	<0.001	ASD vs. ADHD; ASD vs. LD; ASD vs. DCD; ID vs. LD
ZBCS—Personal Strain Scale	8.16 (6.82)	<0.001	ASD vs. LD; ASD vs. DCD; ID vs. LD
ZBCS—Role Strain Scale	11.79 (8.52)	<0.001	ASD vs. ADHD; ASD vs. LD; ASD vs. DCD; ID vs. LD
ZBCS—Guilty Scale	3.89 (2.77)	<0.001	ASD vs. ADHD; ASD vs. LD; ASD vs. DCD

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
