# Peer review of "Caregivers’ Burden of School-Aged Children with Neurodevelopmental Disorders: Implications for Family-Centred Care"

_brainsci, 2021, doi:10.3390/brainsci11070875_

Round 1

Reviewer 1 Report

Interesting article.  I would particularly welcome the use of these burden scales on non-parental direct support professionals.    With the workforce crisis that we all seem to be facing, these scales might give us some insight into causal factors related to the crisis.    Obviously, not for this study, but maybe a prompt for a next study.

Couple of methodological questions – Do you have reliability data on either of the burden scale?  It would seem that it would be easy to calculate a Cronbach’s alpha for both of them.   I’m not that concerned because they were obviously reliable enough for you to detect significant differences.    Second, what is the correlation between the two burden indices?   I’m curious as to why two measures were used, particularly if they are highly correlated.   If relatively independent, the use of the two measures would seem fine.  

A final thought – please limit the use of acronyms to common terms like IQ and DSM.   As a reader, I kept having to go back to remind myself what the letters referenced.   Their use does save space, but, at least for me, makes the read more difficult.  

I’d welcome the addition of reliability and correlational data, along with the reduction of acronym use.   I’d encourage the authors to consider using these burden scales to examine the workforce crisis among Direct Support Professional.   Thanks for the interesting read.

Reviewer 2 Report

The topic of this study is novel and interesting. The Intrduction is well written, the statistical analyses are appropriate and the discussion as well conclusion paragraphs are clear.  Nothing much to report or point out.
